# Ectoparasites Infestation to Small Ruminants and Practical Attitudes among Farmers toward Acaricides Treatment in Central Region of Java, Indonesia

**DOI:** 10.3390/vetsci11040162

**Published:** 2024-04-03

**Authors:** Titis Insyari’ati, Penny Humaidah Hamid, Endang Tri Rahayu, Diah Lutfiah Sugar, Nadya Nurvita Rahma, Shelly Kusumarini, Heri Kurnianto, April Hari Wardhana

**Affiliations:** 1Department of Animal Science, Sebelas Maret University, Kota Surakarta 57126, Indonesia; titisins@student.uns.ac.id (T.I.); endangtri72@staff.uns.ac.id (E.T.R.); diah_7848@student.uns.ac.id (D.L.S.); nandadancell54@student.uns.ac.id (N.N.R.); 2Department of Parasitology, Faculty of Veterinary Medicine, Brawijaya University, Kota Malang 65151, Indonesia; shellykusuma224@ub.ac.id; 3National Research and Innovation Agency, Bogor 16122, Indonesia; heri.kurnianto@brin.go.id (H.K.); apri019@brin.go.id (A.H.W.)

**Keywords:** small ruminants, ectoparasites, zoonotic, acaricide, regional, Indonesia

## Abstract

**Simple Summary:**

In small ruminant production systems, infestations of ectoparasites in goats and sheep are often underestimated, despite being a common issue. Although the cost of treating ectoparasites in small ruminants may initially seem low, it can accumulate significantly due to repeated treatment use and the eventual requirement of higher doses. Our research focused on investigating ectoparasite infestations in traditional farming practices in Central Java, Indonesia. Our findings revealed that 97.85% (637 out of 651, 95% CI: 96.42–98.82%) of the small ruminants examined showed multiple ectoparasite infestations, with various clinical manifestations. Furthermore, surveys conducted with local farmers identified a need for greater knowledge about ectoparasites and their life cycles, which may account for the current inadequate control measures and lack of awareness surrounding these issues. Our findings emphasize the need for increased attention to ectoparasite infestations in farming and the transformation of practical knowledge into active control strategies, given the high occurrence rate of, and public health risks associated with, zoonotic pathogens.

**Abstract:**

Ectoparasite infestations are one of the major problems affecting goat and sheep farming. Disease resulting from these infestations can cause changes in physical appearance, such as severe lesions on the skin, and economic consequences in the form of significantly reduced selling prices. This study aimed to determine the prevalence of ectoparasites in the Boyolali district, Central Java, Indonesia. A total of 651 sheep and goats were surveyed in this study. The parasites were collected via skin scraping, twister, or manually from clinically infected goats and sheep in traditional farms. All of the ectoparasites collected were successfully identified. The prevalence of ectoparasites in ruminants in Boyolali was 97.8% (637/651). The species make-up was as follows: *Bovicola caprae* 97.8% (637/651), *Linognathus africanus* 39% (254/651), *Haemaphysalis bispinosa* 3.5% (23/651), *Ctenocephalides* spp. 0.2% (1/651), and *Sarcoptes scabiei* 5.2% (34/651). The predilection sites were in the face, ear, and leg areas, and in the axillary, dorsal, abdomen, and scrotum regions of the surveyed animals. An evaluation of farmers’ attitudes to ectoparasites was performed using a questionnaire. The findings of this study imply that animals in the investigated area are highly exposed to ectoparasite infestations. Given the importance of ectoparasites in both livestock and human communities, specifically in the health domain, more research into appropriate control strategies is necessary.

## 1. Introduction

Indonesia has the largest small ruminant’s population in Southeast Asia. Among the busiest animal farming regions in Indonesia is Boyolali, within the central region of Java. It was reported that the population of goats and sheep in the Boyolali District in 2020 was 156,962, comprising a total of 55,435 sheep and 101,527 goats [1]. This number represents close to the largest regional population of small ruminants in Java, which 20% of rural areas rear for [2]. Small ruminants, such as goats and sheep, play an important role in the livelihood of rural farming. Small ruminants contribute to household income, especially during Moslem festivities [3]. These animals are raised under conventional management [4] through holding pens or a combination of grazing and confinement systems [2]. Lower amounts of feed and water are needed due to their small body size [5]. Javanese thin-tailed sheep and Kacang goats are the most common native breeds raised by the majority of farmers in Java for the purposes of meat and manure production [6]. Various crossbreds have also been introduced via the animal-upgrading program of the Indonesian government. For example, local sheep were mated with Merino and Texel sheep, whilst the goats were crossed with Ettawa, Saanen, and Boer goats. However, small ruminants farmed in Java are susceptible to diseases [6], which acts as a constraint for production efficiency.

Several studies have revealed that the ectoparasites of small ruminants contribute to extreme skin damage, and heavy infestations can lead to severe anemia [7]. Skin parasites, including lice, ticks, mites, and fleas, live in stationary or temporary conditions on the host’s skin or skin surface for sustenance, maturation, and multiplication. The salivary secretion of some tick species has also been reported to cause various allergies, paralysis, and toxicoses. Their bites can cause mechanical damage, irritation, inflammation, and hypersensitivity, as well as transmit diseases, such as babesiosis, theileriosis, ehrlichiosis, and anaplasmosis [8,9]. Moreover, one study revealed frequent transmission of arthropod borne-pathogens by parasites, such as protozoa, rickettsia, bacteria, and viruses, some of which are zoonotic [10]. The hard tick species are obligate blood-sucking ectoparasites with some pathogenic importance, especially in subtropical and tropical regions [11]. Previous studies have shown that the emergence and reemergence of various tick-borne diseases cause serious public health problems. Until a decade ago, ticks from the *Hyalomma, Rhipicephalus,* and *Haemaphysalis* species, along with tick-borne diseases (TTBDs) such as Crimean-Congo hemorrhagic fever (CCHF), as well as Q fever, received little attention, as most studies were focused on bovines due to their higher economic value. However, there have been recent developments in the socioeconomic significance of small ruminants in food security and poverty alleviation in resource-poor farming communities. This has led to an increase in the attention directed towards a better understanding of TTBDs in sheep and goats [12].

Treatments for ectoparasites include lindane, crotamiton, precipitated sulfur, benzyl benzoate, malathion, phoxim, carbaryl, permethrin, ivermectin, imidacloprid, phenol, and phenylpyrazole [13,14,15,16,17,18,19,20]. However, resistance to these treatments often occurs [21]. Resistance is caused by inappropriate drug prescription, mismatch with the dose prescribed, and no rotation of acaricides used [22,23]. Since the use of acaricides is still the major curative and preventive strategy for ectoparasite infestations [24], careful evaluation of substance usage is necessary. Additionally, the knowledge levels and attitudes of farmers in relation to dealing with acaricides usage and arthropod-borne diseases are also important factors to be elucidated. Although ectoparasites have a significant negative effect on small ruminant farming, there is no information regarding their occurrence and distribution in the central region of Indonesia so far. Therefore, this study aimed to determine the prevalence and associated risk factors of ectoparasites, as well as the perception of famers towards them, in the central region of Indonesia. This baseline information is beneficial for the control and prevention of ectoparasitic infestations in order to avoid economic losses.

## 2. Materials and Methods

### 2.1. Ethical Clearance

The animal experiments related to the observation and skin scraping of goats and sheep in this study were approved by the ethics committee of Universitas Ahmad Dahlan, Yogyakarta with approval no. 022206037.

### 2.2. Animals

The livestock involved in this study were goats and sheep. A total of 651 animals (537 goats and 114 sheep) of different age groups, sexes, and breeds were sourced from 39 traditional farmers and examined. The age of the animals was determined based on the information provided by the farmers and observation of dentition, while assessment of their clinical manifestations was performed visually. The observations were performed throughout the ears, thighs, brisket, ventral abdomen, axillary region, legs, scrotum, dorsal body parts, and face. Palpation across all parts was carried out to check for the possible presence of parasites under the animal hair.

### 2.3. Locations

The area investigated is located between 110°22′–110°50′ east longitude and 7°7′–7°36′ south latitude with an altitude of 75–1500 m above sea level. This places the area of interest near the center of Java Island, Indonesia (Figure 1).

### 2.4. Parasites Collection and Identifications

The ectoparasite samples, i.e., lice, fleas, and ticks were taken directly from the animals by hand [25]. Skin scrapings were performed in animals showing clinical manifestations of infestations. All samples collected were then stored in plastic containers and transported to the UPT of the integrated laboratory at Sebelas Maret University, Surakarta, Indonesia.

Identification of mites was performed by examining the skin scrapings obtained. Skin scrapings from the active mange lesions were preserved in 70% ethanol. The skin debris was then dissolved in a 2% KOH solution [26], and gently burnt in a Bunsen to obtain an aqueous solution. Subsequently, the solution was dropped onto object glass, covered by a cover slip, and observed under a light microscope (Olympus, Tokyo, Japan). The figures of mites were obtained using a microscope camera (Optilab, Bandung, Indonesia). The mites were then identified based on taxonomic morphological keys [27]. Additionally, fresh skin scrapings were processed directly for scanning electron microscopy using JCM 7000 (JEOL, Seoul, Republic of Korea) at PUI Baterai Lithium UNS, Surakarta at high vacuum mode. The tissue sections (obtained from animals sold by farmers during the survey and slaughtered at Ampel abattoir, Boyolali district) were fixed using 10% formalin, and then processed for hematoxylin-eosin staining. The histology studies of ear tissues with infestation lesions were carried out at the Anatomical Pathology Laboratory at the Faculty of Medicine, Gadjah Mada University. Fleas, ticks, and lice samples were soaked in clove oil until the specimens were clear to be identified [28]. Thereafter, the specimens were put onto object glass, and covered by a coverslip. The ectoparasites were permanently fixed using Canada balsam and identified according to taxonomy keys [27].

### 2.5. Farmers Questionnaire

A structured questionnaire was created to collect information on the general characteristics of goat and sheep owners. Furthermore, a questionnaire survey was carried out to obtain information on the sociodemographic profiles of the farmers and assess their awareness of, and control practices against, ectoparasites. Information regarding cures and prevention procedures for infestation, as well as risk factors for disease incidence, were also collected.

### 2.6. Statistical Analysis

The prevalence of ectoparasites was calculated using the formula as follows:

Prevalence (%) = number of infected samples/number of samples tested × 100% [29].

A Microsoft Excel spreadsheet was used for raw data management, while the statistical software SPSS version 26 (IBM, Armonk, NY, USA) was used for data analysis. The association of the prevalence and distribution of ectoparasites with different risk factors, including age, sex, breed, and types of husbandries, was analyzed using a Chi-square test and followed by Fisher’s Exact Test. The differences were considered significant if the *p* value < 0.05, at a 95% confidence interval. The map of the study location was created by Quantum GIS version 3.10 Coruña (OSGeo, Beaverton, OR, USA).

## 3. Results

### 3.1. Results

Rates of lice, mite, tick, and flea infestations in goats were 97.4% (523/537), 6.3% (34/537), 4.3% (23/537), and 0.2% (1/537), respectively (Table 1). All sheep examined were infested with lice. In this study, ectoparasite infestations occurred in all investigated breeds (Figure 2), i.e., Boer 100% (2/2), Jawarandu 96.3% (361/375), Boer × Jawarandu 100% (81/81), Ettawa crossbreed 100% (37/37), Saanen 100% (6/6), Sapera 100% (36/36), Merino 100% (12/12), and thin-tailed sheep 100% (102/102) (Table 2). Clinical manifestations of infestation were observed in various body regions of the animals (Figure 3A), i.e., the face (Figure 3B), ears (Figure 3C,E), axillary region (Figure 3D), legs, scrotum, brisket (Figure 3F), and dorsal parts of animals (Figure 3G,H). Single ectoparasite infection rates were observed to be 88% (473/537) on goats and 100% (114/114) on sheep. The remainder of the examined goats presented concomitant infections by two (8%, 42/537) or three ectoparasites (1%, 8/537). The ectoparasites were distributed across the nine sub-districts investigated, namely Boyolali, Cepogo, Ngemplak, Karanggede, Klego, Musuk, Mojosongo, Nogosari, and Teras (Table 3).

### 3.2. Lice Infestation

Lice infestations were observed in 97.4% (523/537) of goats and 100% (114/114) of sheep (Table 1). Lice were observed on the dorsal, face, thigh, and brisket areas (Figure 3). In terms of breed, the prevalence of lice was 100% (2/2) in the Boer breed, 96.3% (361/375) in the Jawarandu breed, 100% (81/81) in the Jawarandu cross Boer breed, 100% (37/37) in the Ettawa cross breed, 100% (6/6) in the Saanen breed, 100% (36/36) in the Sapera breed, 100% (12/12) in the Merino breed, and 100% (102/102) in thin-tailed sheep (Table 2).

### 3.3. Mange Infestation

Mites diagnosed from skin scrapings were of the sarcoptic mange variety, i.e., *Sarcoptes scabiei*. The *S. scabiei* mites had an oval, nearly round, body shape, four pairs of legs (for adults, the third and fourth pair do not develop and attach to the body), and the body diagonals of specimens were measured to be ±181.74 µm × 265.04 µm (Figure 4A,B). In this study, the epidermal surfaces of ruminants infested with mites showed intensely dry crust. A tissue section of the ear from an infected goat was characterized by mites surrounded by hyper keratinoid stratum corneum, as shown in Figure 4C,D. The mites were observed at various stages and sizes, which represented developmental stages in the goat host. Furthermore, *S. scabiei* burrowed areas were observed, such as a “serpentine-canal” to lay eggs in the skin of the goat. In this study, scanning electron microscopy (SEM) also found *Chorioptes* sp. in the sarcoptic lesion of the goat sample (Figure 5).

Mange infestations were observed in 6.3% (34/537) of goats, while no infections were seen to have occurred in sheep (Table 1). Mange lesions were observed on the face, ears, axillary region, legs, scrotum, brisket, and dorsal areas pf affected goats (Figure 3). Amongst the breeds, the prevalence was 50% (1/2) for Boer, 7.7% (29/375) for Jawarandu, 2.5% (2/81) for Jawarandu cross Boer, and 5.6% (2/36) for Sapera (Table 2). The prevalence of mites in animals less than 2 years old was 5.4% (20/372) and 5% (14/279) in animals more than 2 years old. The prevalence of mites in males was 9.4% (6/64), and in females, it was 4.8% (28/586). The prevalence of mites was 4.7% (26/549) in flock-type housing, and 7.8% (8/102) in individual-type housing. The prevalence of mites in grazing-type animals was 5.3% (2/38), and the prevalence in non-grazing animals was also 5.3% (32/613) (Table 4).

### 3.4. Tick Infestation

Tick infestations were observed in 4.3% (23/537) of goats. No infections were observed in sheep (Table 1). Ticks were observed on the ears, face, and tail of infected goats (Figure 3). Based on the breed, the prevalence was 6.1% (23/375) for Jawarandu (Table 2). The prevalence of ticks in animals less than 2 years old was 2.7% (10/372), and for animals more than 2 years old the prevalence was 4.7% (13/279). The prevalence of ticks was 3.1% (17/549) in flock-type housing, and 5.9% (6/102) in individual-type housing. No infections were observed in grazing-type animals, and the prevalence of infection in non-grazing animals was 3.8% (23/613). The prevalence of ticks in males was 7.8% (5/64), while 3.1% (18/586) of females were affected (Table 4).

### 3.5. Flea Infestation

Flea infestations was observed in 0.2% (1/537) of goats. No infections were observed in sheep (Table 1). Fleas were observed on the dorsal body parts and ventral abdomen o the affected animal (Figure 3). The prevalence for the Jawarandu breed was 0.3% (1/375) (Table 2). There was no recorded prevalence of fleas in flock-type housing, and 1% (1/102) of the total animals in individual-type housing were affected. The single flea infestation was observed in an animal less than 2 years of age, i.e., the prevalence for this age group was 0.3% (1/372); no infection occurred in animals more than 2 years old. The prevalence of fleas in males was 1.6% (1/64), while no infection occurred in females. No infection occurred in grazing-type animals; the sole infection was in a non-grazing animal, representing a prevalence of 0.2% (1/613) (Table 4).

### 3.6. Farmers’ Knowledge and Practices to Ectoparasites Cases in Boyolali

The respondents to the survey were residents of the Boyolali district, and were all engaged in small ruminant farming as their main or side occupation. The majority of the small ruminants’ owners, which served as respondents, were male, representing 89.7% (35/39) of the sample, as shown in Table 5. Furthermore, 64.1% (25/39) of them were more than 46 years old, while those aged 36–45 represented 23.1% (9/39) of the respondents, and 12.8% (5/39) were in the 18–35 age range. All of the respondents had also obtained formal education to various degrees, i.e., elementary school 23.1% (9/39), junior high school 2.6% (1/39), senior high school 48.7% (19/39), and university level 25.6% (10/39). A total of 64.1% (25/39) of the respondents owned 1–10 ruminants, while 23.1% (9/39) and 12.8% (5/39) owned 11–50 and 51–100, respectively. Interviews showed that only 20.5% (8/39) of the respondents knew that some ectoparasites were zoonotic and or disease-carrying vectors. It was discovered through the interviews that the infected animals were treated in various ways, including using topical ointments in 2.6% (1/39) of cases, spray in 5.1% (2/39) of cases, and injection by a local veterinarian in 41.1% (16/39) of cases. Repetitive treatments of less than four injections were found in 68.75% (11/16) of those treated, while an injection frequency of more than four times was found in 31.25% (5/16). Furthermore, 33.3% (1/3) of the farmers performed self-treatment using pour-on acaricide on less than four occasions, whilst this treatment was performed more than four times by 66.7% of the farmers that used it (2/3).

## 4. Discussion

Ectoparasite infestations are often neglected. Nevertheless, ectoparasite infestations arguably should garner a significant amount of attention. Ectoparasites are well known as vectors of various diseases due to their zoonotic properties, which can lead to bigger losses [30]. In Indonesia, some infectious pathogens, i.e., *Babesia* sp. [31,32], *Theileria* sp. [33], *Anaplasma* sp. [34], *Coxiella burnetii* [35], *Scabies scabiei* [36,37], *Borrelia burgdorferi* [38], *Ephemerovirus* [39], *Yersinia pestis* [40], *Chrysomya bezziana* [41], and *Demodex* sp. [42] are known to be caused and/or transmitted by ectoparasites. In this study, 97.8% (637/651) of the small ruminants investigated were infested by one or more ectoparasites. This prevalence is higher than in Nganjuk, East Java [43] and Deli Serdang, North Sumatera [44], according to the only other related investigations reported in Indonesia to date. Given the infestation prevalence exceeding 90%, our result is higher compared to the prevalence of ectoparasite infestations observed in studies conducted in other countries and continents, i.e., in Malaysia [25], Pakistan [45], India [46], Iran [47], Nigeria [48], Ethiopia [49,50], Africa [51], the UK [52], Brazil [53], and Iraq [54].

We identified two different types of lice in this study. *B. caprae* is known as a chewing lice, while *L. africanus* is known as a blood-sucking lice [55]. Infestations of *B. caprae* and *L. africanus* have been reported at various rates in Indonesia [43], Malaysia [25], India [56], Egypt [57], Libya [58], and Brazil [59]. Lice having the highest prevalence among the ectoparasite types may indicate that they are easily transmitted within populations. Goat and sheep lice infestations were almost all subclinical and not visually observable since their predilection sites were covered by hair. Our investigated animals were grouped in very close contact. This condition presumably serves as a risk factor for high transmission and thus a high prevalence rate within small ruminants in the investigated area [55]. Lice infestations in small ruminants can cause clinical manifestations in the forms of irritation through scratching, damaged skin through excessive rubbing, and ultimately alopecia [60,61,62]. Moreover, lice infestation could be responsible for general production loss and could also act as a vector for zoonotic diseases [63].

In this study, mite infestation was the second most prevalent (5.2%, 34/651) type of ectoparasitic infection, with higher number of cases observed in goats than in sheep. The diagnosis of infected goats and sheep was supported by the histologic findings reflecting the presence of burrowing mites of different sizes, as well as eggs in the stratum corneum of the epidermis. It is known that in order for infection and impairment of the host’s immune system to occur, *S. scabiei* must penetrate the skin and burrow to form a pathognomonic lesion [64]. Their saliva has potent digestive enzymes that dissolve skin tissues and facilitate the penetration of the deeper parts of the stratum corneum or the superficial layers of skin. Hyperkeratosis and crust formations were also clearly observed in the affected animals, which are typical signs of scabiosis lesions [65,66]. Chorioptic mange was morphologically identified according to the presence of large caruncles and short pedicels. Chorioptic mange is one of the non-burrowing mites, and it is less pathogenic compared to *S. scabiei* [67,68]. During the collection of skin scrapings to find *S. scabiei*, this ectoparasite was also isolated from the surface of the skin. These results indicate coinfection with burrowing and non-burrowing mites in the same lesion.

The tick species infesting goats and sheep in this study was solely identified as *Haemaphysalis bispinosa.* Previously, *H. bispinosa* has been reported in surrounding countries in South Asia, i.e., Malaysia [69], Singapore [70], Bangladesh, Pakistan, India [71,72], Sri Lanka [73], and Laos [74]. *H. bispinosa* is a parthenogenic species that can reproduce without mating with a male [75]. Therefore, these ticks, once introduced, can easily inhabit and establish new colonies in an area. *H. bispinosa* is noteworthy as other ticks have an environmental stadium which requires attention in control strategies. The farmers in the investigated area did not have sufficient knowledge of the non-host cycle of these ticks (Table 5); they instead concentrate on treating infested small ruminants with acaricides.

Regarding the flea infestation observed in this study, the discovery of the *Ctenocephalides* sp. represents the first record of fleas infesting goats and sheep in Indonesia. Infestations in small ruminants relating to this species have been previously reported in Ethiopia [76], Nigeria [77], Africa [78], and Turkey [79]. The typical hosts of *Ctenocephalides* sp. are dogs, cats, and rodents [80]. Goats and sheep are known as incidental hosts. The infestation occurred only in one goat and was possibly due to the translocation of fleas from cats and dogs observed in the environment around the farm when the sampling was performed. The *Ctenocephalides* sp. is known as a vector for various diseases, i.e., *Yersinia pestis* [81], *Rickettsia typhi* [82], *Francisella tularensis*, and *Listeria monocytogenes* [80].

Almost all of the surveyed farmers had some knowledge about ectoparasites, but experienced difficulties differentiating between lice, mites, ticks, and fleas. Observed ectoparasites were predominantly restricted to the facial and dorsal body parts, and thus implied other predilection sites were lacking attention. Descriptively, more than seventy-nine percent of the farmers had no awareness about the zoonotic aspects of ectoparasites, nor the fact that arthropod-borne diseases can harm human health. This lack of awareness extended to even participants with high levels of education (university or college).

Our analyses showed no significant difference between education levels and ectoparasite infestations on the farms (*p* > 0.05). This observation may be explained by the fact that the educational backgrounds of the farmers were potentially unrelated to animal health and sciences in general. People in the area who work in livestock husbandry were not always educated in animal sciences, which could have otherwise led to improved production and health management. Farmers’ perceptions of the zoonotic properties of ectoparasites were not significantly associated with parasitosis in the region, as determined by univariate analyses (*p* > 0.05). Therefore, the lack of understanding in this area increases the likelihood that the farmers will potentially become infected with zoonotic parasites, i.e., scabietic mange and/or neglected pathogens spilled over from ectoparasites.

Furthermore, most of the farmers had performed acaricide treatments through the use of pour on, injection, spray, and topical ointment methods of application. However, the active compounds included in the acaricides commonly used were not explored in detail within this study. Active substances for treatment of animal ectoparasites in the Indonesian market include ivermectin by injection, sulfur and benzyl benzoate in topical ointments, as well as permethrin and piperonyl butoxide in pour-on treatments. Some traditional farmers often use a small amount of kerosene to bathe affected ruminants, but this poses a toxication risk to the animals. Repetitive treatments administered more than four times in less than 2 months occurred in 31.25% of cases where they were administered by local veterinarian injections, and more than 66.7% when farmers medicated the ruminants themselves. Local veterinarians stated that they often use two to three times higher than normal dosages to treat scabietic lesions in goats and sheep (personal communication). According to the result of univariate analyses, neither self-treatments such as spraying, pouring, or bathing, nor veterinary services, had a significant impact on ectoparasite infestations (*p* > 0.05). These medications were primarily administered within the host-dependent stadium, with no concern given regarding non-host stages, such as egg and larval stages, in the environment. This knowledge led to infectious stadia, which may infect the host after the antiparasitic drug’s onset, but at an insufficient concentration level. An understanding of the life cycle of ectoparasites can lead to the use of multiple antiparasitic medications, resulting in resistance to certain active antiparasitic. Consequently, the lack of effectiveness of medications used in the studied regions may indicate the development of resistances. There was also no correlation between the ages of the farmers and observed ectoparasite infestations (*p* > 0.05). This indicates that the vast majority of farmers in these age ranges, <35 to >46 years old, have similar experience in health practices. The dissemination of new knowledge did not appear to be kept up to date by either the relevant agencies or the initiatives of the farmers themselves.

Altogether, the lack of knowledge about ectoparasites, including their life stages and curative treatments, along with mixed perceptions about their importance, have had a major influence on attitudes towards ectoparasites and awareness of the diseases they cause. These anthropogenic factors unavoidably affect the endemicity and epidemiology of ectoparasites. In addition to the fact that our reports indicated a high prevalence of arthropod-borne diseases in the area, we emphasized the importance of the extensions programs to increase farmers’ knowledge of control and prevention of ectoparasites.

## 5. Conclusions

Ectoparasites are significantly prevalent in goats and sheep within the central region of Java, Indonesia. These infestations lead to disruptions in animal production and may present public health concerns owing to the zoonotic properties of the ectoparasites, such as scabiosis, and the pathogens they harbor. Given the frequent close interactions between livestock and humans in the area under investigation, further studies on epidemiology, pathogen transmission by ectoparasites, and integrated control strategies are deemed essential. It is important to increase farmers’ awareness of appropriate farming practices and to impart knowledge regarding the life cycle of these parasites and, therefore, facilitate the development of effective control strategies. Practical medication protocols should also account for dosage rotation and adherence to minimize the development of parasite resistance to effective substances.

## Figures and Tables

**Figure 1 vetsci-11-00162-f001:**
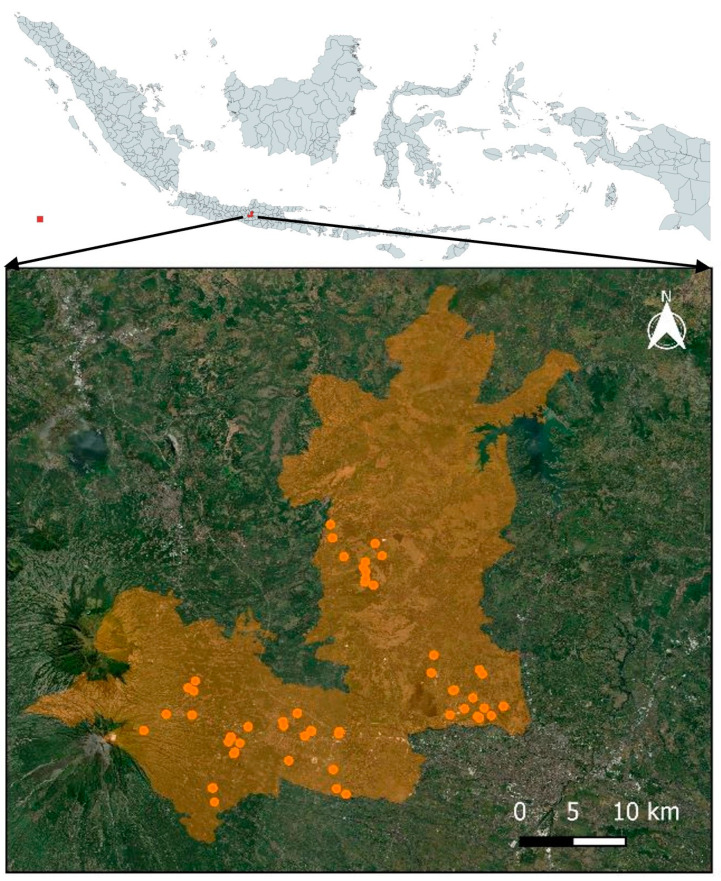
Sampling location of ectoparasites in small ruminants in Boyolali District, Indonesia. The orange dots served as a visual representation of the location of animals’ flocks for the purpose of ectoparasite sampling within the district area (orange shade).

**Figure 2 vetsci-11-00162-f002:**
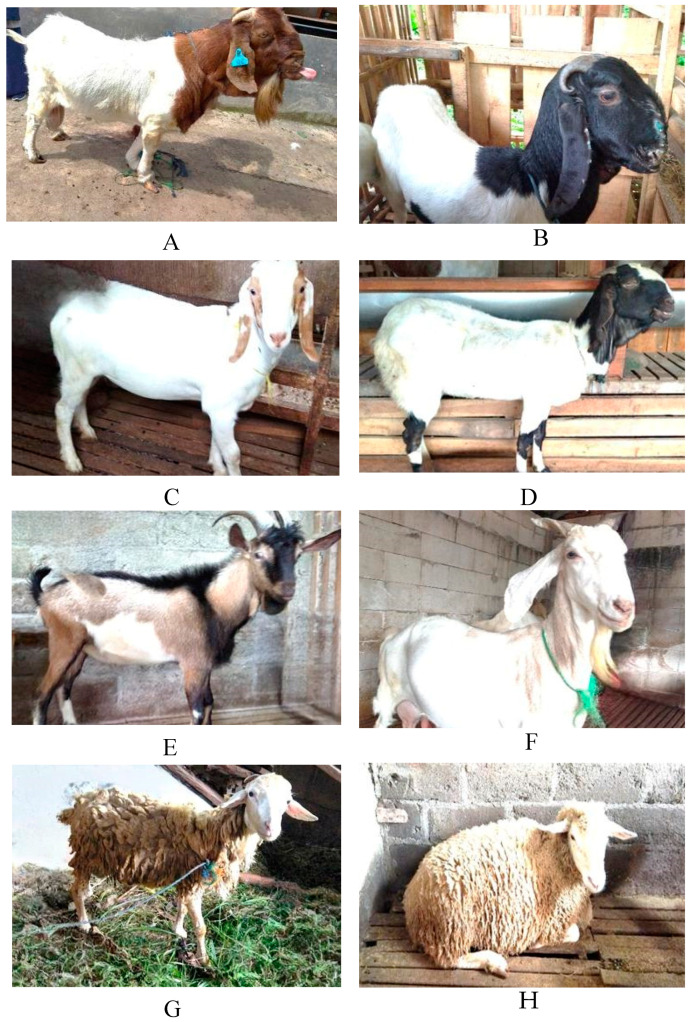
Breeds of surveyed goats and sheep in Boyolali District, Indonesia: (**A**) Boer, (**B**) Ettawa crossbreed, (**C**) Jawarandu, (**D**) Boer × Jawarandu, (**E**) Saanen, (**F**) Sapera, (**G**) Thin-tailed sheep, and (**H**) Merino.

**Figure 3 vetsci-11-00162-f003:**
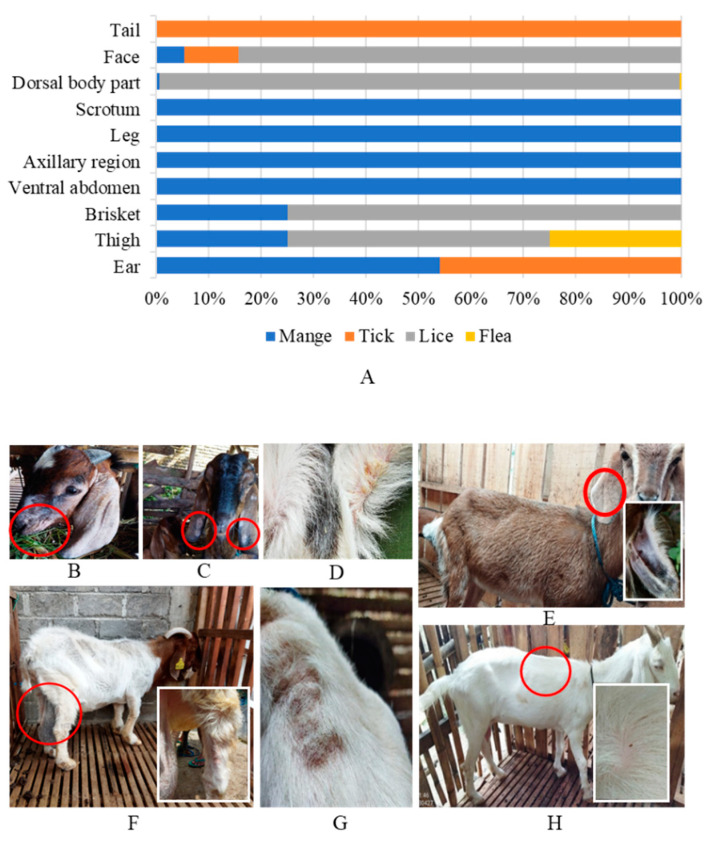
(**A**) Predilection sites of ectoparasites on small ruminants in Boyolali District as indicated by red circles, Indonesia. Lesions observed on (**B**) face, (**C**,**E**) ears, (**D**) the axillary region, (**F**) scrotum, leg, and brisket, and (**G**,**H**) the dorsal region. Skin roughness, alopecia, and hair loss were clearly observed on animals infested with ectoparasites.

**Figure 4 vetsci-11-00162-f004:**
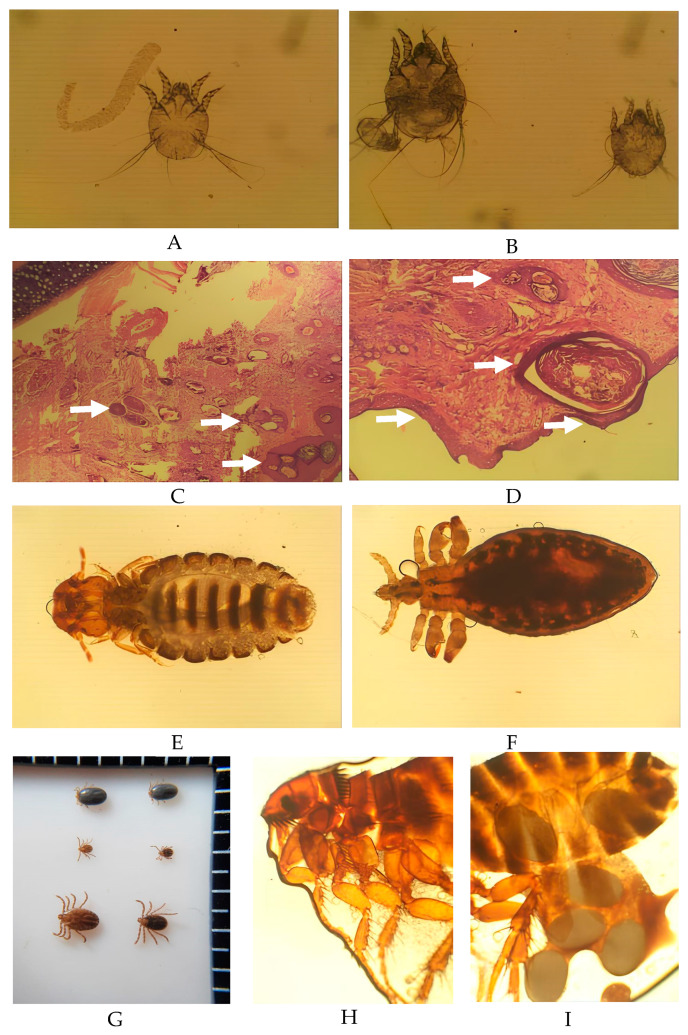
Ectoparasites of small ruminants in Boyolali District, Indonesia. Adult *Scabies scabiei var ovis* (**A**), with egg and larval stages (**B**); hematoxylin eosin stainings of tissue from a goat with severe Scabiotic lesion (**C**,**D**), showing the burrowing mites (arrows): *Bovicola caprae* (**E**), *Linognathus africanus* (**F**), *Haemaphysalis bispinosa* (**G**), and *Ctenocephalides* spp. (**H**,**I**) scale bar 100 μm.

**Figure 5 vetsci-11-00162-f005:**
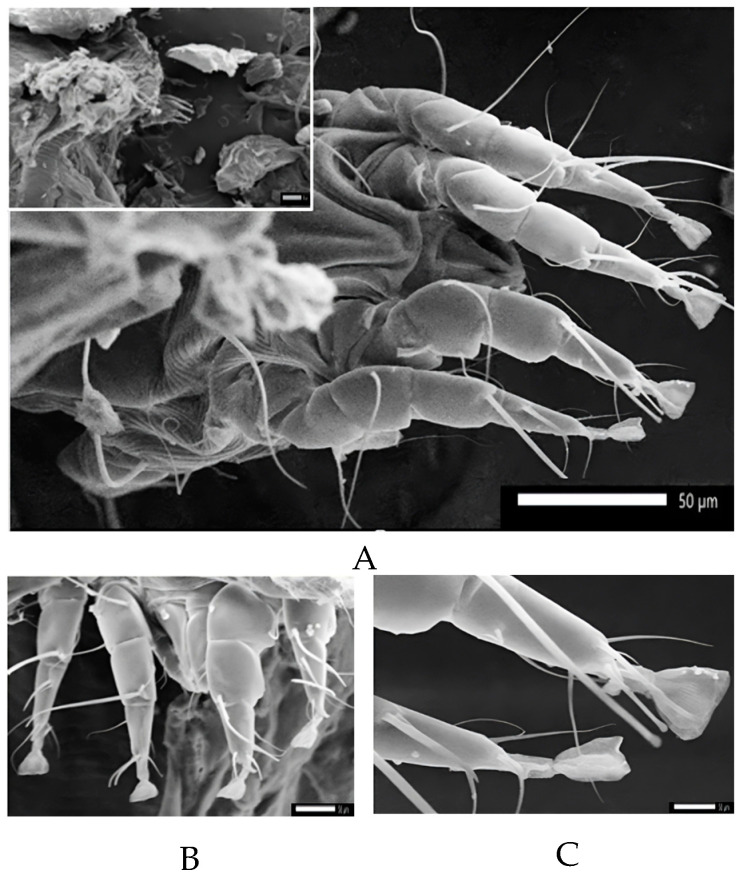
Scanning electron microscopy of a skin scraping from an ovine scabiosis lesion also revealed the presence of *Chorioptes* sp., as indicated by the short pedicel of the pretarsi (**A**,**B**) and the large caruncles (**C**).

**Table 1 vetsci-11-00162-t001:** Prevalence of ectoparasites on goat and sheep in Boyolali District, Indonesia.

Infestations	Goat (*n* = 537)		Sheep (*n* = 114)	
	Prevalence (%)	95% CI	Prevalence (%)	95% CI
Mite	6.30	4.27–8.40	-	-
Tick	4.30	2.57–6.00	-	-
Lice	97.40	96.04–98.74	100.0	100.00
Flea	0.20	0.00–1.03−018–055	-	-
Overall	97.4	96.04–98.74	100.0	100.00

**Table 2 vetsci-11-00162-t002:** Prevalence of ectoparasites according to breed of goat and sheep in Boyolali District, Indonesia.

Breed	*n*	Ectoparasite									
		Mite		Tick		Lice		Flea		Total	
		Positive (*n*)	%	Positive (*n*)	%	Positive (*n*)	%	Positive (*n*)	%	Positive (*n*)	%
Goat											
Boer	2	1	50.0	0	0.0	2	100.0	0	0.0	2	100.0
Jawarandu	375	29	7.7	23	6.1	361	96.3	1	0.3	361	96.3
Boer × Jawarandu	81	2	2.5	0	0.0	81	100.0	0	0.0	81	100.0
Ettawa crossbred	37	0	0.0	0	0.0	37	100.0	0	0.0	37	100.0
Saanen	6	0	0.0	0	0.0	6	100.0	0	0.0	6	100.0
Sapera	36	2	5.6	0	0.0	36	100.0	0	0.0	36	100.0
Total	537	34	6.3	23	4.3	523	97.4	1	0.2	523	97.4
Sheep											
Merino	12	0	0.0	0	0.0	12	100.0	0	0.0	12	100.0
Thin-tailed	102	0	0.0	0	0.0	102	100.0	0	0.0	102	100.0
Total	114	0	0.0	0	0.0	114	100.0	0	0.0	114	100.0
Overall	651	34	5.2	23	3.5	637	97.8	1	0.2	637	97.8

**Table 3 vetsci-11-00162-t003:** Prevalence of ectoparasites according to the study areas in Boyolali District, Indonesia.

Study Area (Subdistrict)	*n*	Ectoparasite									
		Mite		Tick		Lice		Flea		Overall	
		Positive (*n*)	%	Positive (*n*)	%	Positive (*n*)	%	Positive (*n*)	%	Positive (*n*)	%
Boyolali	155	2	1.3	0.0	0.0	155	100.0	1	0.6	155	100.0
Cepogo	30	2	6.7	0	0.0	30	100.0	0	0.0	30	100.0
Ngemplak	63	5	7.9	7	11.1	63	100.0	0	0.0	63	100.0
Karanggede	38	2	5.3	0	0.0	38	100.0	0	0.0	38	100.0
Klego	7	3	42.9	7	100.0	7	100.0	0	0.0	7	100.0
Musuk	31	6	19.4	9	29.0	17	54.8	0	0.0	17	54.8
Mojosongo	111	3	2.7	0	0.0	111	100.0	0	0.0	111	100.0
Nogosari	117	9	7.7	0	0.0	117	100.0	0	0.0	117	100.0
Teras	99	2	2.0	0	0.0	99	100.0	0	0.0	99	100.0
Overall	651	34	5.2	23	3.5	637	97.8	1	0.2	637	97.8

**Table 4 vetsci-11-00162-t004:** Prevalence of ectoparasites in small ruminants according to their age, sex, housing, and management systems in Boyolali District, Indonesia.

Ectoparasites	Age		*p* Value	Sex		*p* Value	Housing Type		*p* Value	Management System		*p* Value
	<2 Years	>2 Years		Male	Female		Individu	Flock		Grazing	Non-Grazing	
	(*n* = 372)	(*n* = 279)		(*n* = 64)	(*n* = 586)		(*n* = 102)	(*n* = 549)		(*n* = 38)	(*n* = 613)	
Mite	20 (5.4%)	14 (5.0%)	>0.05	6 (9.4%)	28 (4.8%)	>0.05	8 (7.8%)	26 (4.7%)	>0.05	2 (5.3%)	32 (5.3%)	>0.05
Tick	10 (2.7%)	13 (4.7%)	>0.05	5 (7.8%)	18 (3.1)	>0.05	6 (5.9%)	17 (3.1%)	>0.05	0 (0.0%)	23 (3.8%)	>0.05
Lice	364 (97.8%)	273 (97.8%)	>0.05	64 (100.0%)	572 (97.6%)	>0.05	102 (100.0%)	535 (97.4%)	>0.05	38 (100.0%)	599 (97.7%)	>0.05
Flea	1 (0.3%)	0 (0.0%)	>0.05	1 (1.6%)	0 (0.0%)	>0.05	1 (1.0%)	0 (0.0%)	>0.05	0 (0.0%)	1 (0.2%)	>0.05
Overall	364 (97.8%)	273 (97.8%)	>0.05	64 (100.0%)	572 (97.6%)	>0.05	102 (100.0%)	535 (97.4%)	>0.05	38 (100.0%)	599 (97.7%)	>0.05

**Table 5 vetsci-11-00162-t005:** Farmers’ perception of ectoparasites and associated treatments in small ruminants.

Variable	Category	No. of Response (%)
Gender	Male	89.7
	Female	10.3
Ages	18–35	12.8
	36–45	23.1
	>46	64.1
Education	No formal education	0
	Elementary school (6 years)	23.1
	Junior high school (9 years)	2.6
	Senior high school (12 years)	48.7
	High education (university level)	25.6
Total small ruminants	1–10	64.1
	11–50	23.1
	51–100	12.8
Ectoparasites knowledge	Flea	2.6
	Lice	66.7
	Tick	15.4
	Mite	43.6
	Don’t know	20.5
Predilection site of ectoparasites mostly observed	Ear	7.7
	Thigh	0.5
	Brisket	0.6
	Ventral abdomen	0.3
	Axilary region	0.3
	Leg	0.6
	Scrotum	0.1
	Dorsal body part	76.2
	Face	25.5
	Tail	0.3
Knowledge of zoonotic potential	Know	20.5
	Don’t know	79.5
Acaricides treatment	Yes, pour on powder	0
	Yes, topical ointment	2.6
	Yes, spray	5.1
	Yes, injection by local veterinarian	41.1
Repetitive treatments by injection	<4 times scheduled injection by local veterinarian	68.75
	>4 times scheduled injection by local veterinarian	31.25
Repetitive treatments by pour on acaricides	<4 times self-treatment	33.3
	>4 times self-treatment	66.7

## Data Availability

Data is contained within the article.

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
