# Peer review of "Ectoparasites Infestation to Small Ruminants and Practical Attitudes among Farmers toward Acaricides Treatment in Central Region of Java, Indonesia"

_vetsci, 2024, doi:10.3390/vetsci11040162_

Round 1

Reviewer 1 Report

Comments and Suggestions for Authors

The authors describe ectoparasite infestations on small ruminant farms in Indonesia. The document is clear and well written. A few remarks/comments nonetheless:

A. Materials and methods

When was this study carried out: duration and seasons (rainy/dry?)

Figure 1: to be changed by also locating the area in relation to Java

line 93: where are the results?

Farmers questionnaire: more details requested. How were the farms chosen? There is a distribution of 7 to 155 farms per district, why? 

3. Results and Discussion: these two parts should be separated, first the results and then the discussion.

Table 1 ff: instead of Mite, ...Indicate the Latin names of the species, identify the columns better, there is possible confusion.

Figure 2: there is no difference between photos A and B, correct? Prefer to put the letter from A to H before the breed name. 

Is there a presentation by breed concerning the presence of parasites? It would be interesting to discuss their respective susceptibility (imported, local, crossbred).

Figure 3: improve the figure, especially with another code (black and white if possible). The photos are not very informative.

Farmer's knowledge: details of the various treatments would be welcome, products, correct use, etc. This information should also be included in the questionnaire.

Line 104: Is there an explanation for this high prevalence? 

Recommendations should be detailed at the end of the discussion or with the conclusion.

4. Discussion = Conclusion (and recommendations)? To be developed, particularly the recommendations and outlook. 

Author Response

Revisions letter to manuscript ID: vetsci-2908503

Dear Editor,

We would like to express our gratitude to the reviewers for their time and constructive feedback on our manuscript. We have incorporated nearly all of the suggestions and revised the manuscript in line with the reviewers' comments, including English editing. We appreciate the guidance provided by the reviewers in improving the manuscript.

We kindly request that the manuscript now be considered for acceptance in your scientific journal.

Best Regards,

Penny Hamid

Response letter to Reviewer 1.

The authors describe ectoparasite infestations on small ruminant farms in Indonesia. The document is clear and well written. A few remarks/comments nonetheless:

  1. Materials and methods
  2. When was this study carried out: duration and seasons (rainy/dry?)

The study was conducted in transition time of dry and rainy seasons (August-October)

  1. Figure 1: to be changed by also locating the area in relation to Java.

Thank you. We revised it based on the suggestion (Figure 1 of revised manuscript).

  1. line 93: where are the results?

All the goat and sheep were presented in Results section (please refer to Table 2).

  1. Farmers questionnaire: more details requested. How were the farms chosen? There is a distribution of 7 to 155 farms per district, why?

The farms were chosen random. Distributions are presented in Figure 1 (yellow dots on the map). We performed random sampling to estimate prevalence.

  1. Results and Discussion: these two parts should be separated, first the results and then the discussion.

Thank you. We revised it as suggestion.

  1. Table 1 ff: instead of Mite, ...Indicate the Latin names of the species, identify the columns better, there is possible confusion.

We presented common names since there are more than 1 species in some ectoparasites.

  1. Figure 2: there is no difference between photos A and B, correct? Prefer to put the letter from A to H before the breed’s name.

A and B were different goat as seen with their colour. We changed to put letter first as suggested.

  1. Is there a presentation by breed concerning the presence of parasites? It would be interesting to discuss their respective susceptibility (imported, local, crossbred).

We presented breed type on Table 2.

  1. Figure 3: improve the figure, especially with another code (black and white if possible). The photos are not very informative.

Figure 3 presents the occurrence of each ectoparasite. To ease differentiation, we used four different colours on the Figure.

  1. Farmer's knowledge: details of the various treatments would be welcome, products, correct use, etc. This information should also be included in the questionnaire.

The respondents were farmers and not local veterinarians. They were lacking comprehensive knowledge about the specific drugs they use. Consequently, the relevant information is limited.

  1. Line 104: Is there an explanation for this high prevalence? 

It may be due to high number of small-ruminants population in the area, ecology and awareness of the farmers. However, we did not performed analysis on the factors may involve in the current manuscript.

  1. Recommendations should be detailed at the end of the discussion or with the conclusion.

Thank you. We rewrote this section in the revised manuscript.

  1. Discussion = Conclusion (and recommendations)? To be developed, particularly the recommendations and outlook. 

As answered in question no. 12 and conclusion

Reviewer 2 Report

Comments and Suggestions for Authors

The title corresponds with the content of the article.

Number of animals checked for the study is adequate. The Authors collected ectoparasites and surveyed the farmers, which gives important practical data.

Abstract well summarize the text, just first lines (10-11) are not necessary.

Please skip very old references, as the data can not be compared. The number of non actual references is quite big.

The manuscript still needs some work to improve its value.

Overall opinion is good.

To the keywords list reviewer would add the region of the study.

Introduction contains important statistical data of small ruminants population, which gives good background for the study. Importance of vector - borne diseases as public health treat is underlined. The Introduction part clearly presents the gap in the knowledge and goal of the study.

line 84 please do not begin with Table. And please chceck the numeration of the Tables (double 1 -misleading)

There is no reference in the text to figure 1, figures represent high added value to the text

line 93 "different" is not enough information; those variables are influencing on the statistics.

The farmers voluntarily agreed for participation in the study?

line 102 sealed containers? how the samples were described?more details needed, who was responsible for parasites collection & identification?

line 112 - there is no information about ear tissue collection in previous parts

In the M&M part dedicated to statistical analysis please add p value.

lines 78-93 survey results should be more broadly presented

the rest part of the Results paragraph is satisfactory.

line 130-131 and corresponding fig should be moved to the results part, SEM not mentioned in M&M part.

line 155 the companion animals presence should be included in data collection and analyzed as variable statistically 

line 157 did Authors make any microbiological tests?

discussion is quite well prepared, with practical conclusion at the end. Big part of the discussion is dedicated to the survey, which was barely described in the results part. Part of the discussion dedicated to ectoparasites could be more dynamic in discussion with other papers. The public health treat in underlined once again.

line 204 conclusions not discussion

conclusions are too general must be re-written

Comments on the Quality of English Language

no serious faults in the language found by the reviewer who is not native speaker

Author Response

Revisions letter to manuscript ID: vetsci-2908503

Dear Editor,

We would like to express our gratitude to the reviewers for their time and constructive feedback on our manuscript. We have incorporated nearly all of the suggestions and revised the manuscript in line with the reviewers' comments, including English editing. We appreciate the guidance provided by the reviewers in improving the manuscript.

We kindly request that the manuscript now be considered for acceptance in your scientific journal.

Best Regards,

Penny Hamid

Response letter to Reviewer 2.

  1. The title corresponds with the content of the article.
  2. Number of animals checked for the study is adequate. The Authors collected ectoparasites and surveyed the farmers, which gives important practical data.
  3. Abstract well summarize the text, just first lines (10-11) are not necessary.

Thank you. We eliminated line 10-11 in revised version.

  1. Please skip very old references, as the data can not be compared. The number of non actual references is quite big.
  2. The manuscript still needs some work to improve its value.
  3. Overall opinion is good.
  4. To the keywords list reviewer would add the region of the study.

We added keywords: regional; Indonesia (line 24 in revised manuscript).

  1. Introduction contains important statistical data of small ruminants’ population, which gives good background for the study. Importance of vector - borne diseases as public health treat is underlined. The Introduction part clearly presents the gap in the knowledge and goal of the study.
  2. line 84 please do not begin with Table. And please check the numeration of the Tables (double 1 -misleading)

Many thanks for the suggestions. We revised based on suggestions.

  1. There is no reference in the text to figure 1, figures represent high added value to the text

We added in line 95 of revised manuscript.

  1. line 93 "different" is not enough information; those variables are influencing on the statistics.

Detailed informations were provided in Table 4. We stated in line 83 of revised version.

  1. The farmers voluntarily agreed for participation in the study?

Yes

  1. line 102 sealed containers? how the samples were described?more details needed, who was responsible for parasites collection & identification?

Yes. The authors listed.

  1. line 112 - there is no information about ear tissue collection in previous parts

We added in line 113, section material and methods.

  1. In the M&M part dedicated to statistical analysis please add p value.

We added in line 133 (line 135 of revised ver)

  1. lines 78-93 survey results should be more broadly presented

We added it in line 51 of result part (revised version)

  1. the rest part of the Results paragraph is satisfactory.
  2. line 130-131 and corresponding fig should be moved to the results part, SEM not mentioned in M&M part.

Thank you. We removed from discussion to result section as suggested. We added SEM in 112-113, section materials and methods

  1. line 155 the companion animals presence should be included in data collection and analyzed as variable statistically 

We did not perform collection in companion animals. We just saw some pet animals around the farm.

  1. line 157 did Authors make any microbiological tests?

No

  1. discussion is quite well prepared, with practical conclusion at the end. Big part of the discussion is dedicated to the survey, which was barely described in the results part. Part of the discussion dedicated to ectoparasites could be more dynamic in discussion with other papers. The public health treat in underlined once again.
  2. line 204 conclusions not discussion

Thank you

  1. conclusions are too general must be re-written

We rewritten in revised version
